# Healthcare providers' knowledge on sickle cell disease and its management: A pre- and post-training test evaluation outcome

Vivian Paintsil[1,2]*, Evans Xorse Amuzu[2], Eunice Ahmed[3],
Yaa Gyamfua Oppong-Mensah[2], Lesley Osei[4], Bernice Eklu[2],
Suraj Yawnumah Abubakar[2], Lawrence Osei-Tutu[2], Daniel Ansong[1,2],
Alex Osei-Akoto[1,2], Fred Stephen Sarfo[5,6]

1 Department of Child Health, School of Medical Sciences, Kwame Nkrumah University of Science and Technology, Kumasi, Ghana, 2 Sickle Cell Unit, Directorate of Child Health, Komfo Anokye Teaching Hospital, Kumasi, Ghana, 3 Directorate of Laboratory Medicine, Komfo Anokye Teaching Hospital, Kumasi, Ghana, 4 Directorate of Transfusion Medicine, Komfo Anokye Teaching Hospital, Kumasi, Ghana, 5 Department of Internal Medicine, School of Medical Sciences, Kwame Nkrumah University of Science and Technology, Kumasi, Ghana, 6 Directorate of Internal Medicine, Komfo Anokye Teaching Hospital, Kumasi, Ghana

* vivpee@yahoo.com

## Abstract

In resource-limited settings in Africa, which harbour the greatest burden of Sickle Cell Disease (SCD) globally, poor care outcomes are driven in part, by a lack of trained healthcare providers (HCP) and an absence of context-specific treatment guidelines appropriate to the level of healthcare facility. The study aimed to evaluate the impact of a structured training program on HCP's knowledge of SCD in Ghana. This was prospective cross-sectional study involving HCPs from 46 health facilities from 4 out of 16 regions in Ghana. A curriculum and standard slides were developed by SCD experts based on the Sickle Pan African Research Consortium (SPARCO)Standards of care for Sickle Cell Disease in sub-Saharan Africa clinical recommendations. A full-day workshop highlighting the general overview of SCD, diagnosis, health maintenance, acute and chronic complications was then organized. A pre-training test and a post-training test immediately after the workshop were administered and analyzed. A total of 543 HCPs were trained, mostly from primary level facilities (77.7%). The average number of years working with SCD patients was 5 years (Range: < 1–20 years). Most (93%) HCPs had experience with SCD patients but only 43% reported using a form of guideline for the care of SCD patients. The average score in the pre-training test was 8.4/20 (SD:3.3) increasing to 13.1/20(SD:3.6) in the post-training test, (p-value <0.01). The average proportion of persons indicating a correct answer for a question was 50% at the pre-training test increasing to approximately 69% in the post-test, (p-value <0.01). The knowledge of HCPs about SCD and its management was generally low but improved significantly after the standardized training.

**Data availability statement:** All relevant data for this study are publicly available from the Zenodo repository (https://doi.org/10.5281/zenodo.16099761).

**Funding:** Funding was received from Pfizer specialties limited to conduct the trainings.

**Competing interests:** The authors have declared that no competing interests exist.

Further studies are required to assess the impact of HCP training on health outcomes of SCD in resource limited settings.

## Introduction

Sickle cell disease (SCD) is the commonest clinically significant haemoglobinopathy worldwide [1] and considered a public health problem [2]. It predominantly affects blacks and its clinical manifestations starts in the first year of life. In Ghana, about 1.8% of our newborn population have SCD with predominantly SCD-SS and SCD-SC genotypes [3]. Patients with SCD can develop both acute and chronic complications if they are not given any comprehensive treatment [4]. Grosse et al estimated that about 50–90% of patients under 5 years of age with SCD will die if no treatment is initiated [5]. To prevent patients from developing these complications, they are enrolled into dedicated SCD clinics to benefit from heath maintenance interventions. This includes the provision of evidence-based interventions namely penicillin prophylaxis, folic acid supplementation, hydroxyurea for patients, psychosocial support, required vaccinations and regular screening for organ damage including eye, brain, kidney and the heart [6]. Aside routine visits, patients with SCD can present with various complications that needs to be managed adequately by healthcare workers. The commonest amongst these are Vaso-Occlusive pain episodes (VOPE) which present with severe pain that if not adequately managed can lead to chronic pain. Other complications like Acute Chest syndrome (ACS) can also occur during an episode of VOPE and can lead to death [4]. Other common complications that can occur include stroke which causes disability for the patients and requires rehabilitation by the healthcare workers.

The management of SCD requires a multidisciplinary approach that requires physicians and surgeons with various subspecialties, nurses and other allied healthcare professionals [6]. Their input in the management of a patient with SCD is critical for increased survival of the patient. This requires that healthcare professionals are well vexed in the management of SCD wherever they find themselves.

The World Health Organization (WHO) has recommended that management for SCD should be incorporated into primary healthcare, hence all healthcare workers should be able to have adequate knowledge and skills which pertains to diagnosing, and managing SCD patients [6]. Insufficient knowledge will undoubtedly lead to misdiagnosis, increasing morbidity and mortality in this population. To make sure that SCD patients everywhere are treated the same way, there is the provision of standard of care guidelines which is utilized by the healthcare workers. Previous evaluation of knowledge in other studies showed a suboptimal level of knowledge. In Nigeria, only 37.9% of primary healthcare workers had good knowledge about SCD [7] while in DRC, 44% followed any guideline or recommendations for the management of VOC and pain management [8]. In Tanzania, only 25.1% had good knowledge about SCD [9].

The Sickle Pan African Research Consortium (SPARCO) has developed and published recommendations for the management of SCD which is available for use in

Sub-Saharan Africa [10]. Ghana is one of the SPARCO sites and the Komfo Anokye Teaching Hospital, a tertiary hospital with a dedicated SCD clinic, where these guidelines form the standard of care for the management of patients. However, to facilitate the diffusion of knowledge, skill and competencies in SCD management to HCP working in lower cadres of healthcare delivery, we undertook a study to assess the knowledge of healthcare workers in 4 regions about SCD management and the impact of the structured training program on their knowledge.

## Methods

### Study design

This was a prospective cross-sectional study carried out between August 2023 – May 2024 in 4 regions of Ghana. Facilitators were paediatric and adult haematologists at the Komfo Anokye Teaching hospital who run the Kumasi Centre for Sickle Cell Disease. The curriculum and standard PowerPoint slides were drawn from the SPARCO Standards of Care Guidelines [10]. The training was a whole day's workshop that dealt with overview and pathophysiology of SCD, health maintenance for SCD, Acute complications (VOPE, ACS, Stroke, Anaemia) and Chronic complications (Retinopathy, leg ulcers, nephropathy) and slides found as S4 appendix. Discussion forums were further used to build on the knowledge acquired by the HCWs.

To recruit participants, we initially looked out for facilities that had SCD clinics running, then subsequently invited facilities who referred patients with SCD to our facility and other bigger facilities whom we expected to refer patients but that was not occurring were also targeted. An invitation was purposively sent to the management of these facilities, and they were requested to nominate persons who attend to SCD patients (doctors, nurses, physician assistants, biomedical scientists) to attend the workshops.

After registration, participants were requested to answer a questionnaire. This questionnaire comprised two sections: (1) sociodemographic data: sex, education, city of residence, and professional category, type of health service and the number of years working in healthcare services; and (2) a 20-question knowledge test. The 20-question knowledge test was drawn from the SPARCO Standards of Care Guidelines [10] by the facilitators, ensuring that it covered everything that would be covered during the workshop and found as S1 Appendix. The questions were also reviewed by other experts in SCD for their relevance, clarity and comprehensiveness of the questions. The questions were subsequently pretested to ensure it would test the knowledge appropriately, covered the learning objectives and were clear and understandable. The study team incorporated the necessary modifications into the questions before finalizing it. These same questions were used for both pre-test and post-test. Participants were asked to use their initials so we could map the initial and post-test to ascertain the degree of increase in learning.

### Inclusion and exclusion criteria

All Healthcare workers who attended the workshop were eligible to be part. No one was excluded.

### Ethical considerations

This study was performed under the SPARCO project with approval by the KNUST Committee for Human Research and Publication Ethics with approval references CHRPE/AP/088/23 and CHRPE/AP/273/24.

### Data collection

Participants completed a pre-test before the training and completed a post-test immediately after the training.

Knowledge of participants was assessed using a set of 20 questions which covered SCD diagnosis, management, risk screening, complications and cure.

## Data analysis

Data was analysed using STATA 17.0. categorical variables are presented as frequencies and percentages while numeric variables are presented as the mean and the standard deviation or the median and inter quartile range depending on the normality of its distribution.

Test questions were assigned a score of 1 for a correct answer and 0 for a wrong answer. The overall score was further categorized into three levels of knowledge as elucidated by Alzahrani et al using the Bloom's cut-off point [11]. Scores 16–20(80–100%) were labelled as high level of knowledge, 12–15(60–79%), moderate knowledge, while scores <12 are labelled as having a low knowledge of SCD.

Pairwise correlation was used to find the strength and direction of association the participants' age, years of working and years of experience with SCD with the pre-test score. A linear regression model was then used to model the significant associations.

The effect of the training was derived from a subset of the data with matched pre- and post- test scores. A two-sample paired student t-test with equal variances was used to test the difference in the scores pre and post the training change.

## Results

In all, a total of 543 healthcare workers participated in the training session and completed either the pre-training test, and/or the post-training test. Out of this number 376 participated in the pre-training test and provided their demographic information. Approximately 69% of the trainees completed at least the post-training test. Approximately 39% (210/543) of these participants participated in both the pre -training and the post- training tests as shown in Fig 1.

Most of participants were from the primary health care facilities and majority of those who completed the post-test (81.9%) were from this category. Nurses formed the majority (50% vs 53%) of those who participated in both the pre-test and post-test respectively. Majority (61.4%) of participants were not using any guidelines for managing SCD as seen in Table 1.

Facility level and occupation were found to be statistically significant predictors of knowledge level at pre-training. Compared to persons working in primary health facilities, persons working in secondary level facilities on average had 1.4 points higher knowledge scores in their pre-training test scores. Compared to doctors, nurses had 4 points lower knowledge scores in their pre-training scores. (Table 2)

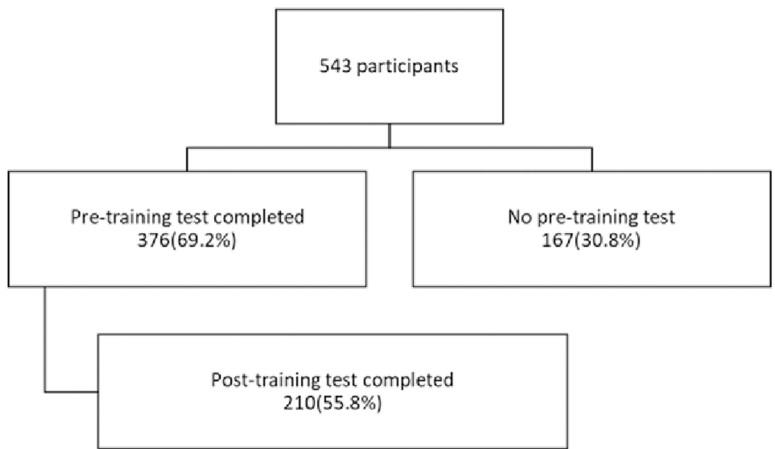

**Fig 1. Flow diagram of participants recruitment.**

**Table 1. Demographic characteristics of participants.**

| Variable | Pre-test n = 376 Frequency, n (%) | Post-test n = 210 Frequency, n (%) |
|---|---|---|
| **Facility Level** | | |
| Primary | 292(77.7) | 172(81.9) |
| Secondary | 22(5.9) | 14(6.7) |
| Tertiary | 62(16.5) | 24(11.4) |
| **Sex** | | |
| Female | 232(61.7) | 125(59.5) |
| Male | 144(38.3) | 85(40.5) |
| **Age** Mean (SD) | 33.5(5.9) | 33.0(5.6) |
| **Occupation** | | |
| Doctors | 92(24.5) | 41(19.5) |
| Nurses | 188(50.0) | 112(53.3) |
| Others | 96(25.5) | 57(27.1) |
| **Work experience(years)** Mean (SD) | 7.1(5.4) | 7.0(5.1) |
| **Experience with SCD** | | |
| No | 23(6.4) | 14(6.9) |
| Yes | 339(93.7) | 188(93.1) |
| **SCD Working experience (Years)** Mean (SD) | 4.8(4.4) | 4.6(4.1) |
| **Use of SCD guidelines** | | |
| No | 231(61.4) | 115(59.6) |
| Yes | 145(38.6) | 78(40.41) |

**Table 2. Linear regression table of factors predictive of knowledge at pretest.**

| Variable | Crude (95% CI) | p-value | Adjusted (95% CI) | p-value |
|---|---|---|---|---|
| **Age** | 0.02 (−0.03- 0.08) | 0.411 | 0.02(−0.07-0.12) | 0.631 |
| **Working time** | 0.01(−0.05- 0.08) | 0.675 | 0.02(−0.01-0.14) | 0.741 |
| **Working with SCD** | 0.09(0.01- 0.17) | *0.027* | 0.02(−0.08-0.12) | 0.656 |
| **Sex** | | | | |
| Female | Ref | Ref | Ref | Ref |
| Male | 0.23(−0.46- 0.92) | 0.509 | −0.05(−0.76-0.65) | 0.881 |
| **Facility level** | | | | |
| Primary | Ref | Ref | Ref | Ref |
| Secondary | 1.26 (−0.16 −2.68) | 0.081 | 1.44(0.01-2.78) | *0.036* |
| Tertiary | 1.24(0.34-2.14) | *0.007* | 0.40(−0.59-1.38) | 0.427 |
| **Occupation** | | | | |
| Doctors | Ref | Ref | Ref | Ref |
| Nurses | −4.06(−4.77-−3.35) | *<0.001* | −4.15(−4.98—3.32) | *<0.001* |
| Others | −3.74(−4.55-−2.93) | *<0.001* | −3.52(−4.50-−2.55) | *<0.001* |
| **Experience with SCD** | | | | |
| No | Ref | Ref | Ref | Ref |
| Yes | 1.18(0.06- 2.30) | *0.038* | −0.18(−1.73- 1.37) | 0.816 |

## Impact of training

Questions on the unrecognized presentation of SCD, common retinopathy genotype, penicillin v substitute in the event of penicillin allergy, infant diagnosis method, and medication not required in steady state had over 30% more correct answers in the post-training test than in the pre-training test. The average score was higher in the post-training test 13.1(SD 3.6) than in the pretest 8.5(SD 3.3). (Table 3)

Table 4 showed that there was a 4.6-point increase in the mean scores at pre-training compared pretest to post-training scores test of participants. This difference was found to be statistically significant.

Mean pretest scores and post test scores were lowest in participants from primary level facilities as seen in Fig 2. The minimum post-test score of participants emanating from secondary and tertiary facilities were at least equal to the median pretest score.

In Table 5, there was a significantly improved knowledge levels especially among doctors with a shift to the high post-test score. Nurses and other professionals showed a statistically significant but more modest improvement in knowledge.

The proportion of participants failing the test reduced from 65% at pre-training to 21% post-training. This change was found to be statistically significant. Over 50% of those who failed in the pre-training test, passed the post-training test.

## Discussion

### Sociodemographic characteristics of participants

The patient recruitment flow diagram in Fig 1 shows a clear overview of the selection and completion of the pre-test and post-test. About 70% took part in the pre-test whiles only 55.8% took part in the post-test. This attrition could be an indication of the fear for assessments by HCP and may affect the generalizability of the result. The study involved a diverse group of HCPs

**Table 3. Pre and post test questions – Impact of training.**

| Pre-training Correct answer distribution (n) | Frequency (%) | Post-training Correct Answer distribution | Frequency (%) | Diff |
|---|---|---|---|---|
| Not recognized clinical presentation(n = 361) | 124(34.35) | Not recognized clinical presentation p(n = 365) | 278(76.16) | 41.81 |
| Non-SCD complication(n = 365) | 143(39.18) | Non SCD complication p(n = 371) | 195(52.56) | 13.38 |
| Genotype for SCD retinopathy (n = 360) | 88(24.44) | Genotype for SCD retinopathy p(n = 370) | 235(63.51) | **39.07** |
| Post splenectomy vaccines (n = 343) | 125(36.44) | Post splenectomy vaccines p(n = 368) | 195(52.99) | 16.55 |
| Hydroxyurea in anemia management(n = 368) | 169(45.92) | Hydroxyurea in anemia management p(n = 369) | 202(54.74) | 8.82 |
| Aplastic crisis complication (n = 357) | 226(63.31) | Aplastic crisis complication p(n = 360) | 247(68.61) | 5.3 |
| Penicillin V substitute (n = 350) | 138(39.43) | Penicillin V substitute p(n = 370) | 302(81.62) | 42.19 |
| SCD diagnostic tests(n = 365) | 63(17.26) | SCD diagnostic tests p(n = 370) | 142(38.38) | 21.12 |
| Infant diagnosis(n = 364) | 104(28.57) | Infant diagnosis p(n = 369) | 251(68.02) | **39.45** |
| Least useful test for Sβthal(n = 366) | 129(35.25) | Least useful test for Sβthal p(n = 365) | 203(55.62) | 20.37 |
| Not required steady state(n = 368) | 222(60.33) | Not required steady state p(n = 369) | 348(94.31) | 33.98 |
| Mental health and psychosocial support (n = 371) | 17(4.58) | Mental health and psychosocial support p(n = 369) | 15(4.07) | −0.51 |
| Non-pharma prevention of crises(n = 368) | 286(77.72) | Non-pharma prevention of crises p(n = 370) | 324(87.57) | 9.85 |
| Not primary ACS management(n = 361) | 151(41.83) | Not primary ACS management p(n = 368) | 249(67.66) | 25.83 |
| SCD potential for cure (n = 364) | 289(79.4) | SCD potential for cure p(n = 368) | 324(88.04) | 8.64 |
| Hydroxyurea (HU) role in management (n = 361) | 109(30.19) | Hydroxyurea role in management p(n = 370) | 162(43.78) | 13.59 |
| Possible ACS presentation in SCD (n = 361) | 215(59.56) | Possible ACS presentation in SCD p(n = 368) | 260(70.65) | 11.09 |
| Sickle Cell Anaemia definition(n = 341) | 246(72.14) | Sickle Cell Anaemia definition p(n = 367) | 274(74.66) | 2.52 |
| Stroke screening methods(n = 350) | 234(66.86) | Stroke screening methods p(n = 370) | 287(77.57) | 10.71 |
| Key principle in SCD management(n = 359) | 238(66.3) | Key principle in SCD management p(n = 366) | 236(64.48) | −1.82 |
| **Score n = 375 mean(sd)** | **8.47(3.3)** | **Score n = 372 mean(sd)** | **13.07(3.61)** | **4.6** |

**Table 4. Paired t test comparison of pretest and post-test scores by occupations.**

| All occupations (n=210) | mean(95%CI) | t | p-value |
|---|---|---|---|
| Post test score | 12.9(12.3--13.4) | 21.22 | <0.0001 |
| Pre-test score | 8.2(7.8--8.7) | | |
| Diff | 4.6(4.2--5.1) | | |
| **Doctors (n=41)** | | | |
| Post test score | 16.5(15.6--17.3) | 14.1 | <0.0001 |
| Pre-test score | 11.5(10.7--12.3) | | |
| Diff | 5(4.3--5.7) | | |
| **Nurses (n=112)** | | | |
| Post test score | 11.7(11.1--12.3) | 13.4 | <0.0001 |
| Pre-test score | 7.4(6.9--7.9) | | |
| Diff | 4.3(3.7--5) | | |
| **Others (n=57)** | | | |
| Post test score | 12.6(11.5--13.7) | 11.7 | <0.0001 |
| Pre-test score | 7.6(6.7--8.5) | | |
| Diff | 5(4.1--5.8) | | |

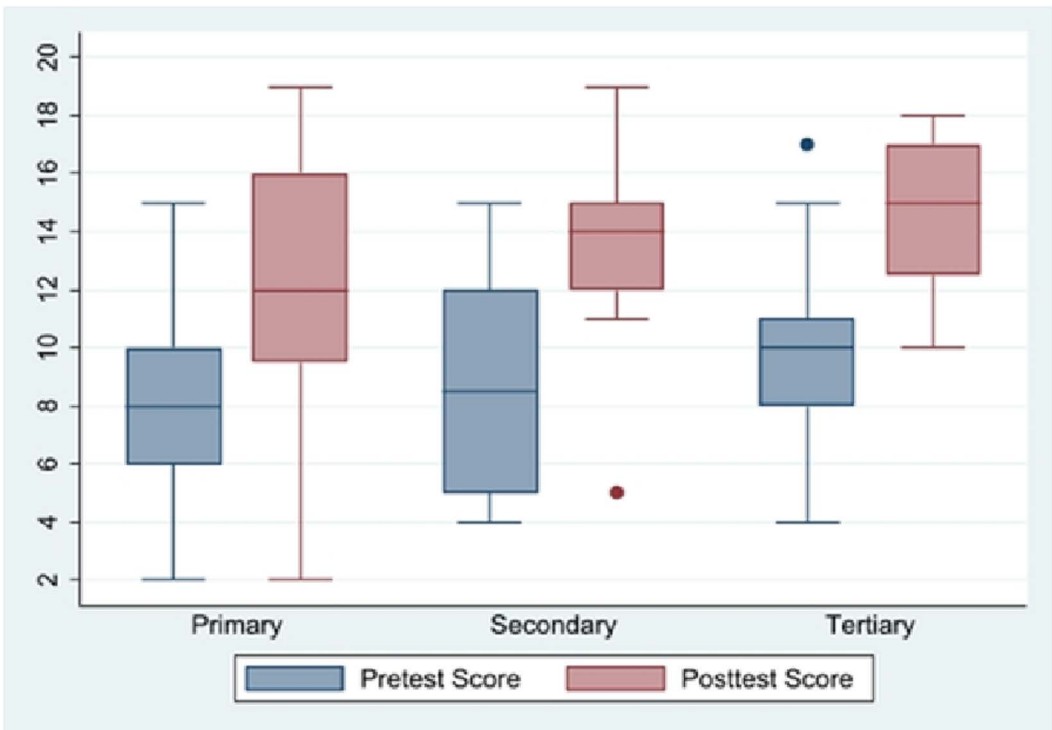

**Fig 2. Boxplot comparing pretest and post-test scores in different referral level facilities.**

involved in the care of SCD patients, including doctors, nurses, biomedical laboratory scientists and pharmacists as seen in Table 1. Participants were largely nurses', and this is not surprising as nurses form the largest proportion of the healthcare workforce in Ghana. Our finding of more females than males contradicts findings in a study in Democratic Republic of Congo

**Table 5. Association of pre-and post- test scores by occupations.**

| Doctors | Post test knowledge level | | | | |
|---|---|---|---|---|---|
| Pre-test knowledge level | Low | Moderate | High | Total | x² |
| Low | 2(100) | 7(100) | 12(37.5) | 21(51.22) | 0.005 |
| Moderate | 0(0) | 0(0) | 19(59.38) | 19(46.34) | |
| High | 0(0) | 0(0) | 1(3.13) | 1(2.44) | |
| Total | 2(4.88) | 7(17.07) | 32(78.05) | 41(100) | |
| Nurses | | | | | |
| Low | 50(98.04) | 44(97.78) | 13(81.25) | 107(95.54) | 0.029 |
| Moderate | 1(1.96) | 1(2.22) | 3(18.75) | 5(4.46) | |
| Total | 51(45.54) | 45(40.18) | 16(14.29) | 112(100) | |
| Others | | | | | |
| Low | 24(100) | 14(87.5) | 13(76.47) | 51(89.47) | 0.031 |
| Moderate | 0(0) | 2(12.5) | 4(23.53) | 6(10.53) | |
| Total | 24(42.11) | 16(28.07) | 17(29.82) | 57(100) | |
| All professions | | | | | |
| Low | 76(98.7) | 65(95.6) | 38(8.5) | 179(85.2) | <0.001 |
| Moderate | 1(1.3) | 3(4.41) | 26(40.0) | 30(14.29) | |
| High | 0(0.0) | 0(0.0) | 1(1.54) | 1(0.48) | |
| Total | 77(36.7) | 68(32.4) | 65(30.9) | 210(100.0) | |

(DRC) where their medical workforce were predominantly males [12]. Our results are comparable to the health workforce of Ghana where around 60% are females [13]. Most participants worked at primary healthcare facilities as seen in Table 2 and this was a purposeful move to involve more Primary Health Care (PHC) centres to benefit from the training. The World Health Organization (WHO) [6] in its strategic framework for SCD has emphasized the need to integrate SCD care into the primary health care system so that no matter where patients find themselves, they would have access to good care.

### Baseline knowledge about SCD

The Healthcare Workers showed a low level of knowledge at pre-training (Table 3) which is comparable with other African countries where knowledge assessment in HCWs showed a poor knowledge base [7,9]. This was not the case for a trainer-of-trainer (ToT) workshop where participants had a good baseline knowledge with the lowest score being 13 out of 20 [14]. The best of HCWs is largely chosen for the ToT and so it was likely that they were familiar with SCD and its management. In contrast to what is reported in our study, Diniz et al from the USA showed that over two-thirds of healthcare workers had good knowledge on SCD [15]. For patient centred care and good patient outcome, a good knowledge about SCD is required. Our results indicate a gap that underscores the need for continuous training on SCD.

The questions with the lowest scores were those on psychosocial support and diagnostic tests for SCD. This is worrying as making a diagnosis of SCD is one of the most essential things to do for patient outcome. Only 17.6% of participants knew how to diagnose SCD patients correctly. This is lower as compared to other studies in DRC where about 26.6% and 48.8% of participants had knowledge about the Emmel test and Hb Electrophoresis respectively. The newborn screening (NBS) program for SCD has been ongoing in Ghana since 1995 and it was expected that HCPs have good knowledge about the program but only 28.75% knew about NBS. Newborn screening for SCD is recommended to identify newborns suffering from SCD before the onset of symptoms, to prevent infectious complications and VOC and to reduce the risk of mortality. This requires the institutionalization of continuous training for HCPs especially as WHO has recommended NBS for SCD. Healthcare workers however had good knowledge about complications and management of SCD.

Participants who had worked previously with SCD patients had a significantly higher pre-training test score as compared to those who had not worked with SCD patients (Table 2). This is comparable with a study in Kinshasa where nurses with previous experience with SCD had a better knowledge [12]. This higher pre-training test score could be as a result of attendance in previous workshops on SCD. It could also be because of accumulated field experience on SCD.

Participants from the secondary and tertiary referral level had a significantly higher score at pre-training compared to those from the primary levels as seen in Fig 2. This could indicate disparities in access to prior training or SCD educational resources. Doctors performed significantly better in the pre-test compared to nurses and other allied health professionals with nurses scoring an average of 4 points lower than doctors. It brings to bear the need for a targeted training according to their professional roles.

The study also highlights a significant gap in the uptake of clinical guidelines for the management of SCD with only 38.6% of participants reporting use of guidelines in their clinical practice as seen in Table 1. In spite of the availability of evidence-based recommendations such as the National Heart, Lung, and Blood Institute (NHLBI), SPARCO guidelines and nationally with the standard treatment guidelines, the integration into routine care remains suboptimal. This may impair the health outcomes of patients with SCD. Potential barriers to guideline uptake could be limited awareness and accessibility of the guidelines. Other systemic constraints like inadequate treatment infrastructure and medication stock-out may discourage adherence to these guidelines even when available. This training is thus important for HCP to be aware and use the guidelines for better patient outcome.

### Impact of training

Results from this project showed a significant impact of the training in improving the knowledge about SCD among HCWs. The pre and post-test analysis shows an increase in knowledge scores depicted by a mean score improvement of 4.6 points ($p < 0.001$) shown in Table 3. The questions within the domains related to unrecognized presentation of SCD, commonest phenotype for retinopathy, Pen V substitutes in the event of penicillin allergy and methods for infant diagnosis showed the greatest improvement with >30% increases (Table 3).

Participants from secondary referral level facilities showed significant improvement with an adjusted score of 1.44 points compared to the primary level facilities (Fig 2). Those from primary level facilities consistently had the lowest median pretest and post test scores which indicates the need to provide more sustained training given that the significant proportion of SCD care is delivered at primary level healthcare facilities in low-income settings. The minimum post-test scores for secondary and tertiary facilities equalled or exceeded the median pretest scores which showed a more robust baseline knowledge and better post training performance. Occupational disparities persisted even after training as shown in Tables 4 and 5 which emphasizes the need for the training to be tailored to the specific roles and responsibilities of the different cadres of health workers.

Despite these overall gains, minimal improvement or slight decline was seen in the question about psychosocial support as part of treatment and key principles of management. It was however unclear why HCPs rather got these questions wrong in the post-training test. It could be that the content was not clear during the presentation, or the test questions were confusing and will need to be reviewed again.

Overall, there was a reduction in the proportion of participants who had low knowledge about the test from 65% pre-training to 21% post-training which further highlighted the effectiveness of the training. While we demonstrated an improvement in healthcare worker's knowledge of SCD following training, it is important to acknowledge that the ultimate measure of impact lies in patient-care outcomes. Improved knowledge should ideally translate into better clinical decision-making, timely interventions and a better quality of care for patients with SCD.

### Conclusion

This training significantly improved knowledge about SCD among HCPs with the greatest gains in clinical management. However, disparities in knowledge across facility levels and occupations highlight the need for targeted role-specific

training especially for primary level facilities to address the persistent knowledge gaps. There will also be the need to incorporate SCD management in all the curriculum for the health training institutions to improve their baseline knowledge about SCD. It also highlights the need to have SCD management guidelines available for HCPs in the management of patients for a better outcome.

### Limitations and future research

The post-test was conducted immediately after the training which does not account for long term retention of the knowledge. There was attrition in the proportions of participants who voluntarily completed pre-test and post-tests out of the total sample of participants who attended the workshops. Participation in the workshop based on nomination from purposively selected facilities which could introduce some selection bias of the participants. Also, the lack of a validated knowledge test could be a potential limitation.

Future studies should include follow-up assessments to evaluate long term knowledge gain and also evaluate whether the training intervention result in measurable improvements in patient care outcomes such as improved adherence to use of guidelines and enhanced patient satisfaction. Also, validating the assessment tool using psychometric properties like internal consistency should be considered for future studies.

### Supporting information

**S1 Appendix.  Test Questionnaire.**
(DOCX)

**S2 Appendix.  SPARCO II 2023–2024 Renewal.**
(PDF)

**S3 Appendix.  SPARCO II Approval 2024–2025.**
(PDF)

**S4 Appendix.  SOC Training Slides.**
(PDF)

### Acknowledgments

We acknowledge the support of the various heads of health facilities for their collaboration and support in organizing the workshops. We also acknowledge the support of Priscilla Agyeibea Awuku and Tony Boakye in data acquisition and entry.

### Author contributions

**Conceptualization:** Vivian Paintsil, Evans Xorse Amuzu.

**Data curation:** Vivian Paintsil, Evans Xorse Amuzu, Eunice Ahmed, Yaa Gyamfua Oppong-Mensah, Lesley Osei, Bernice Eklu, Suraj Yawnumah Abubakar, Lawrence Osei-Tutu, Daniel Ansong, Alex Osei-Akoto, Fred Stephen Sarfo.

**Formal analysis:** Vivian Paintsil, Evans Xorse Amuzu, Bernice Eklu, Daniel Ansong.

**Funding acquisition:** Vivian Paintsil, Evans Xorse Amuzu.

**Investigation:** Vivian Paintsil, Evans Xorse Amuzu, Eunice Ahmed, Yaa Gyamfua Oppong-Mensah, Lesley Osei, Bernice Eklu, Suraj Yawnumah Abubakar, Lawrence Osei-Tutu, Daniel Ansong, Alex Osei-Akoto, Fred Stephen Sarfo.

**Methodology:** Vivian Paintsil, Evans Xorse Amuzu, Eunice Ahmed, Yaa Gyamfua Oppong-Mensah, Lesley Osei, Bernice Eklu, Suraj Yawnumah Abubakar, Lawrence Osei-Tutu, Daniel Ansong, Alex Osei-Akoto, Fred Stephen Sarfo.

**Project administration:** Vivian Paintsil, Evans Xorse Amuzu, Daniel Ansong.

**Resources:** Vivian Paintsil, Evans Xorse Amuzu, Eunice Ahmed, Yaa Gyamfua Oppong-Mensah, Lesley Osei, Bernice Eklu, Suraj Yawnumah Abubakar, Lawrence Osei-Tutu, Alex Osei-Akoto, Fred Stephen Sarfo.

**Supervision:** Vivian Paintsil, Evans Xorse Amuzu, Daniel Ansong, Alex Osei-Akoto, Fred Stephen Sarfo.

**Validation:** Vivian Paintsil, Evans Xorse Amuzu, Lawrence Osei-Tutu.

**Visualization:** Vivian Paintsil, Evans Xorse Amuzu.

**Writing – original draft:** Vivian Paintsil, Evans Xorse Amuzu.

**Writing – review & editing:** Vivian Paintsil, Evans Xorse Amuzu, Eunice Ahmed, Yaa Gyamfua Oppong-Mensah, Lesley Osei, Bernice Eklu, Suraj Yawnumah Abubakar, Lawrence Osei-Tutu, Daniel Ansong, Alex Osei-Akoto, Fred Stephen Sarfo.

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
