## [Decision Letter · Decision Letter 0]

15 May 2025

Dear Dr. Paintsil,

We look forward to receiving your revised manuscript.

Kind regards,

Santosh L. Saraf

Academic Editor

PLOS ONE

Journal Requirements:

2. Please amend either the title on the online submission form (via Edit Submission) or the title in the manuscript so that they are identical.

4. Please ensure that you refer to Figure 2 in your text as, if accepted, production will need this reference to link the reader to the figure.

5. Please remove all personal information, ensure that the data shared are in accordance with participant consent, and re-upload a fully anonymized data set.

6. We note you have included a table to which you do not refer in the text of your manuscript. Please ensure that you refer to Table 1 in your text; if accepted, production will need this reference to link the reader to the Table.

Reviewers' comments:

Reviewer's Responses to Questions

**Comments to the Author**

1. Is the manuscript technically sound, and do the data support the conclusions?

Reviewer #1: Yes

Reviewer #2: Yes

2. Has the statistical analysis been performed appropriately and rigorously?

Reviewer #1: Yes

Reviewer #2: Yes

3. Have the authors made all data underlying the findings in their manuscript fully available?

Reviewer #1: No

Reviewer #2: Yes

4. Is the manuscript presented in an intelligible fashion and written in standard English?

Reviewer #1: Yes

Reviewer #2: Yes

Reviewer #1: This is an interesting article evaluating the impact of a structured training program on healthcare providers’ knowledge of SCD in Ghana. While educating providers about SCD is critical, this pilot study is somewhat limited and has a few significant biases. However, it is a large-scale educational project with some important characteristics. Below are some of the limitations and possible areas to improve the impact of the manuscript.

Major Concerns:

1) Lack of use of a validated instrument for the knowledge test is a significant concern.

a. Several questions are unrelated to sickle cell disease (e.g., #7); it is unclear what the intent is of these questions that may not be directly related to sickle cell knowledge.

b. Numerous acronyms are present that may be confusing and may limit participants’ ability to respond accurately (e.g., #8,10).

c. It would be helpful to include more background about how these questions were made.

d. Was there any validation of these questions? If not, this is a major limitation.

2) There is likely a large knowledge gap between the different occupations, as is noted in the baseline score, and some of the occupations have a more critical role in the treatment of sickle cell disease; therefore, it would be important to know the differences between the occupations for the following:

a. A breakdown of the different occupations’ completion of the pre- and post-test is needed

b. Table 3 would be important to be broken down by occupation.

c. Also, the pre- and post-test differences (Tables 4 and 5) would be interesting if they were analyzed by occupation.

3) Measuring knowledge before and after a single in-person workshop limits the impact of this manuscript. This pilot may not translate to affecting patient care; however, it is a large-scale training, involving many healthcare professionals in Ghana. The manuscript could also mention the importance of measuring patient-care outcomes after this intervention.

4) There is a notable drop-off from those who participated to those who had a pre- and post-test, 55%.

a. This is a very large limitation and should be noted in the limitations section.

b. It would be important to include a supplementary table describing the differences in demographics, profession, and other components between those who gave that information but did not complete the post-test.

Minor concerns:

1) Including the slides of the workshop would be helpful for others who might want to replicate the work.

2) The score cut-off of 10 is arbitrary. Is there any justification that could make this a reasonable cutoff? If not, consider removing this cutoff and the concept of passing and failing from the manuscript.

Reviewer #2: Comprehensive work and well written. i wonder why the cut offs were 0-10 and 11-20. Perhaps having in 3 groups (poor, fair, great) will give a real idea of the difference between pre and post test in various groups.

**Do you want your identity to be public for this peer review?** For information about this choice, including consent withdrawal, please see our Privacy Policy

Reviewer #1: No

Reviewer #2: **Yes: ** Marwah Farooqui

---

## [Author Response · Author response to Decision Letter 1]

27 Jul 2025

Dear Reviewers and Editor,

Point-to-point response to reviewers’ comments

We would like to thank the reviewers and editor for their critical revision and valuable comments which had a tremendous effect on improving our manuscript. The points brought up by the reviewers have provided us with very important insights, and we are confident that the changes made have improved the quality of our work. Please find below our point-to-point responses to the reviewer’s comments.

We hope you find the amendments satisfactory. We are happy and ready for any further steps or improvement.

Reviewer Comments on Manuscript PONE-D-25-17578

Comment Response

1) Lack of use of a validated instrument for the knowledge test is a significant concern. The instrument was not statistically validated but first reviewed by experts in the field of SCD and pretested across all cadres involved in SCD care before its use. It was designed to test knowledge about the SPARCo Standards of Care Guidelines for SCD management in sub-Saharan Africa which was the source of the curriculum for the workshop. A description of its development and validation is inserted Line 101-109

a. Several questions are unrelated to sickle cell disease (e.g., #7); it is unclear what the intent is of these questions that may not be directly related to sickle cell knowledge. All questions were related to SCD. E.g. #7 is aimed at knowing if healthcare workers know what medication to use for prophylaxis for infections when a patient with SCD is allergic to regular Penicillin V

b. Numerous acronyms are present that may be confusing and may limit participants’ ability to respond accurately (e.g., #8,10). The acronyms have been written in full.

c. It would be helpful to include more background about how these questions were made. A description of its development and pretesting is inserted Line 101-109

d. Was there any validation of these questions? If not, this is a major limitation. The instrument was not statistically validated but pretested across all cadres involved in SCD care before its use. A description of its development and validation is inserted Line 101-109

2) There is likely a large knowledge gap between the different occupations, as is noted in the baseline score, and some of the occupations have a more critical role in the treatment of sickle cell disease; therefore, it would be important to know the differences between the occupations for the following:

a. A breakdown of the different occupations’ completion of the pre- and post-test is needed A description of the participants completing the post test has been added to table 1

b. Table 3 would be important to be broken down by occupation. The difference between the pre-test and post-test by the different occupations has been done and now seen as Table 4

c. Also, the pre- and post-test differences (Tables 4 and 5) would be interesting if they were analyzed by occupation. Analysis by occupation has been done and included

3) Measuring knowledge before and after a single in-person workshop limits the impact of this manuscript. This pilot may not translate to affecting patient care; however, it is a large-scale training, involving many healthcare professionals in Ghana. The manuscript could also

mention the importance of measuring patient-care outcomes after this intervention.

This will be done in future studies

4) There is a notable drop-off from those who participated to those who had a pre- and post-test, 55%.

a. This is a very large limitation and should be noted in the limitations section. This is well noted and has been added as a limitation.

b. It would be important to include a supplementary table describing the differences in demographics, profession, and other components between those who gave that information but did not complete the post-test. Description of persons completing post test added to table 1

Minor concerns:

1) Including the slides of the workshop would be helpful for others who might want to replicate the work. This is well noted. The slides will be available for anyone upon request

2) The score cut-off of 10 is arbitrary. Is there any justification that could make this a reasonable cutoff? If not, consider removing this cutoff and the concept of passing and failing from the manuscript. Categorization has now been changed to using Bloom’s cut-off points as was used in a similar study by Alzahrani et al.

Reviewer #2: Comprehensive work and well written. i wonder why the cut offs were 0-10 and 11-20. Perhaps having in 3 groups (poor, fair, great) will give a real idea of the difference between pre and post test in various groups. Categorization has now been changed to using Bloom’s cut-off points which talks about high level of knowledge, moderate knowledge and low knowledge as used by Alzahrani et al.

Additional considerations:

1. Lack of Long-Term Follow-Up: The post-training assessment was immediate. A follow-up at 3–6 months would help assess retention and sustained impact. A follow-up study is being planned for the future

2. Validation of Assessment Tool: The manuscript does not clarify whether the pre/post-test questions were piloted or validated. Including psychometric properties (e.g., internal consistency) would strengthen the evaluation. A description of its development and validation is inserted Line 101-109

3. Cutoff for Knowledge Scores: Comprehensive work. That said, I wonder why the cutoffs were 0–9 (poor) and 10–20 (good). A three-tier system (e.g., poor, fair, good) may give a more nuanced view of the distribution and changes across knowledge levels. Categorization has now been changed to using Bloom’s cut-off points. This has been described in line 125-129

4. Selection Bias: Since participation was based on invitations, there may be a bias toward more engaged or motivated providers. This should be briefly acknowledged in the limitations. Acknowledged. Line 282-283

5. Participant feedback: Consider including a brief participant feedback component in future work to guide content refinement. Thanks and this is well noted

6. Barriers to implementation: Expand on the discussion of barriers to guideline uptake, especially since only 43% reported using one. This has been discussed and found in lines 235-244

Thank you

Dr Vivian Paintsil

---

## [Decision Letter · Decision Letter 1]

16 Aug 2025

Dear Dr. Paintsil,

We look forward to receiving your revised manuscript.

Kind regards,

Santosh L. Saraf

Academic Editor

PLOS ONE

Journal Requirements:

Reviewers' comments:

Reviewer's Responses to Questions

**Comments to the Author**

Reviewer #1: (No Response)

Reviewer #2: All comments have been addressed

2. Is the manuscript technically sound, and do the data support the conclusions?

Reviewer #1: Yes

Reviewer #2: Yes

3. Has the statistical analysis been performed appropriately and rigorously?

Reviewer #1: Yes

Reviewer #2: Yes

4. Have the authors made all data underlying the findings in their manuscript fully available?

Reviewer #1: No

Reviewer #2: Yes

5. Is the manuscript presented in an intelligible fashion and written in standard English?

Reviewer #1: Yes

Reviewer #2: Yes

Reviewer #1: The authors have done a good job responding to this reviewers comments. A few more comments/clarifications would help improve the impact of this manuscript:

Adding the slides as a supplemental file would be helpful.

There was no description of the validation of the knowledge test in the submission. For example, how was content validity determined? The lack of a validated knowledge test should be included in the limitations.

Adding that the following would be done in future studies in the manuscript as part of this reviewer’s original comments would be helpful: “. The manuscript could also mention the importance of measuring patient-care outcomes after this intervention”

Reviewer #2: changing to low, moderate, and high in the pre and post scores clarifies my understanding of this. also appreciated the breakdown based on profession

**Do you want your identity to be public for this peer review?** For information about this choice, including consent withdrawal, please see our Privacy Policy

Reviewer #1: No

Reviewer #2: No

---

## [Author Response · Author response to Decision Letter 2]

21 Aug 2025

Comments and Response

Comments: Adding the slides as a supplemental file would be helpful.

Response: The slides have been added on as supplementary file

Comments: There was no description of the validation of the knowledge test in the submission. For example, how was content validity determined? The lack of a validated knowledge test should be included in the limitations.

Response: This has been added to the limitations and inserted in lines 289-290.

Also, a recommendation has also been inserted in lines 293-295 for validating the knowledge test

Comments: Adding that the following would be done in future studies in the manuscript as part of this reviewer’s original comments would be helpful: “. The manuscript could also mention the importance of measuring patient-care outcomes after this intervention”

Response: A discussion on this has been inserted in lines 272-275. And it has also been added on in lines 291-293 as a recommendation for future studies.

---

## [Editor Report · Decision Letter 2]

26 Aug 2025

Healthcare providers’ knowledge on Sickle Cell Disease and its management: A pre- and post-training test evaluation outcome

PONE-D-25-17578R2

Dear Dr. Paintsil,

We’re pleased to inform you that your manuscript has been judged scientifically suitable for publication and will be formally accepted for publication once it meets all outstanding technical requirements.

Kind regards,

Santosh L. Saraf

Academic Editor

PLOS ONE
---

## [Editor Report · Acceptance letter]

PONE-D-25-17578R2

PLOS ONE

Dear Dr. Paintsil,

I'm pleased to inform you that your manuscript has been deemed suitable for publication in PLOS ONE. Congratulations! Your manuscript is now being handed over to our production team.

Kind regards,

on behalf of

Dr. Santosh L. Saraf

Academic Editor

PLOS ONE